# BRIEF COMMUNICATION: IMPACTS OF A DEVELOPING POLYNYA OFF COMMONWEALTH BAY, EAST ANTARCTICA, TRIGGERED BY GROUNDING OF ICEBERG B09B

Christopher J. Fogwill[1*], Erik van Sebille[2], Eva A. Cougnon[3,4,5], Chris S.M. Turney[1], Steve R. Rintoul[3,4,5],

Benjamin K. Galton-Fenzi[4,6], Graeme F. Clark[1], E.M. Marzinelli[1], Eleanor B. Rainsley[7], Lionel Carter[8].

[1] Climate Change Research Centre, School of Biological, Earth and Environmental Sciences, UNSW Sydney, Australia

[2] Grantham Institute & Department of Physics, Imperial College London, United Kingdom

[3] Institute for Marine and Antarctic Studies, University of Tasmania, Private Bag 129, Hobart, Tasmania

7001, Australia.

[4] Antarctic Climate & Ecosystems Cooperative Research Centre, University of Tasmania, Private Bag 80, Hobart, Tasmania 7001

[5] Commonwealth Scientific and Industrial Research Organisation, Ocean and Atmospheric Research, Hobart, Australia.

[6] Australian Antarctic Division, Kingston, Tasmania

[7] Wollongong Isotope Geochronology Laboratory, School of Earth and Environmental Sciences, University of Wollongong, Wollongong, Australia.

[8] Antarctic Research Centre, Victoria University of Wellington, New Zealand

*Correspondence to c.fogwill@unsw.edu.au


**Abstract.** The dramatic calving of the Mertz Glacier Tongue in 2010, precipitated by the movement of iceberg B09B, reshaped the oceanographic regime across the Mertz Polynya and Commonwealth Bay, regions where high salinity shelf water (HSSW) – the precursor to Antarctic bottom water (AABW) – is formed. Here we present post-calving observations that suggest that this reconfiguration has driven the development of a new polynya off Commonwealth Bay, where HSSW production continues due to the grounding of B09B. Supported by satellite observations and modelling, our findings demonstrate how local icescape changes may impact the formation of HSSW, with potential implications for large-scale ocean circulation.

## 1. Introduction

The events triggered by the movement of the 97 km long iceberg B09B adjacent to the Mertz Glacier Tongue (MGT) in 2010 precipitated a significant iceberg calving event that was captured in real time from satellite data and shipboard observations (Shadwick et al., 2013). Prior to the calving event, Commonwealth Bay – the site of Sir Douglas Mawson's Australasian Antarctic Expedition (AAE) of 1911-1914 – was usually free of sea ice, owing to the presence of an extensive coastal polynya maintained by strong off-shore katabatic winds sustained by the local ice-sheet topography and the presence of the Mertz Polynya to the east. Historically, newly-formed sea ice has been rapidly transported offshore by these winds; for example, during the original AAE of 1911-1914, the sea ice in Commonwealth Bay was stable enough to walk on for only two days each year (Mawson, 1940). In December 2010, however, the grounding of iceberg B09B in Commonwealth Bay in 2010 changed the local icescape considerably (Shadwick et al., 2013; Lacarra et al., 2014) (Figure 1A). The presence of the grounded iceberg B09B since 2010 has blocked the off-shore transport of sea ice, leading to the build-up of year-round fast-ice up to 3m thick landward of the iceberg (Clark et al., 2015). This transition from an area that was often ice-free to one of continuous fast-ice cover has created a natural experiment into the impacts of fast-ice change on both local biota (Clark et al., 2015) and ocean circulation (Shadwick et al., 2013; Lacarra et al., 2014). The latter

is particularly important given the Adélie-George V Land region is a key region of formation of Antarctic bottom water (AABW; a generic term that encompasses the variable nature of such bottom waters (Orsi et al., 1999; van Wijk and Rintoul, 2014; Nihashi and Ohshima, 2015)). Prior to the calving of the Mertz Glacier, both the Mertz and Commonwealth Bay polynyas were important sources of high salinity shelf

water (HSSW) and dense shelf water (DSW) formation, which are precursors to AABW. As AABW supplies the lower limb of the global thermohaline circulation system (Orsi et al., 1999), changes in the properties or rate of formation of AABW in response to the local icescape can influence the continental shelf sea circulation (e.g. Cougnon, 2016), with widespread consequences for deep ocean circulation and ventilation (Kusahara et al., 2011; Shadwick et al., 2013).

The loss of the 78 km-long Mertz Glacier Tongue in 2010, which had previously reduced westward flow of ice into the Mertz Polynya and Commonwealth Bay, is estimated to have caused a marked impact on sea-ice formation regionally (Tamura et al., 2012; 2016). Furthermore, model studies suggest that this has led to a reduction in HSSW formation in the area (Kusahara et al., 2011), a hypothesis supported by *in situ*

observations in 2011/2012 (Shadwick et al., 2013; Lacarra et al., 2014). Together, these data indicate an abrupt reduction in the salinity and density of shelf water and an increase in carbon uptake in the region of the Mertz Polynya when compared to pre-calving levels. Palaeoceanographic studies suggest that the impacts of MGT calving on AABW formation may be a cyclical process, possibly occurring on centennial timescales (Campagne et al., 2015).


Given that the majority of AABW is formed at a number of key sites around Antarctica (Rintoul et al., 1998; Orsi et al., 1999) – including the Weddell Sea, the Ross Sea, Amery-Shackleton ice shelf, Cape Darnley, Vincennes Bay and Adélie-George V Land – any major long-term circulation change in these regions could have a significant impact on the global climate system. At present the long-term stability of

AABW formation is not fully understood, and it is possible that the rates of AABW production from

regional areas are highly variable both temporally and spatially (Broecker et al., 1998). Therefore, studying the impacts of natural perturbations such as the grounding of B09B can provide insights into the sensitivity of AABW formation to past and future changes in regional icescape.

Here we report new data that provides a snapshot of change in the region of the Mertz Polynya and Commonwealth Bay from *in situ* oceanographic observations from December 2013, the austral summer, which suggests the region is in a process of transition towards a new steady state (Figure 1). To explore the potential future impacts of these changes, we use high-resolution ocean model simulations to examine regional ocean dynamics in two steady states (pre- and post-calving), focussing particularly on changes in
velocity and advection of water masses between the Mertz Polynya and Commonwealth Bay for stable scenarios pre- and post-grounding of B09B.

### 2. *In situ* observations and comparison with past data

We report observations of changes in ocean water properties recorded during December 2013 on the
Australasian Antarctic Expedition 2013-2014 (AAE 2013-2014) from the *MV Akademik Shokalskiy*. A research programme was designed to examine the changes in the region since the Mertz Glacier calving event in 2010, building upon observations from previous research expeditions in the region (Shadwick et al., 2013; Lacarra et al., 2014). To compare the current oceanographic conditions in the region with previous measurements, expendable conductivity temperature and depth probes (XCTDs; model XCTD-1,
Tsurumi-Seiki Co.) were deployed. To demonstrate the reliability of the XCTD data, test casts were assessed against repeat casts using Seabird-SBE37SM microcat CTD calibrated for cold water conditions (see SOM Figure S1). A TSK TS-MK-21 expendable XCTD system was used to gather oceanographic data, which was recorded on a laptop computer. Given the marked expansion of fast-ice in Commonwealth Bay, in some locations XCTDs and microcat were deployed through the fast-ice as well as in open water
from the vessel. Although some deployments were opportunistic, many were repeat casts of previous

stations in Commonwealth Bay and in the Mertz Polynya to allow direct comparison with studies taken during past austral summers (Figure 1).

The XCTD results from December 2013 are divided into three geographic areas to allow comparison with data from previous cruises from the same season (Figure 1 and SOM Figure S2). Salinity and temperature data from the austral summer 2013/14 from northwest of Commonwealth Bay ('Commonwealth Bay NW'), northeast ('Mertz NE') and southwest ('Mertz SW') of the MGT are compared to previous years in Figure 1B, C and D respectively. As salinities and water density vary both spatially and seasonally across the region (Lacarra et al., 2014), here we compare our data to that collected in similar seasons (December/January) and locations (Figure 1, SOM Figure S2).

In Commonwealth Bay NW (Figure 1B) our results show an increased salinity at ~350 m (34.62‰) since the pre-calving values of 2008 (34.55‰). We also find a temperature decrease of ~0.2˚C from -1.7˚C in 2008 to -1.9˚C in 2013. In the Mertz NE region (Figure 1C), we measure the salinity at ~550 m to be 34.7‰, higher than the post-calving salinity low in 2012 of ~34.58‰, and similar to those immediately post-calving (2011) and pre-calving (2008). The water is also colder, and importantly, shows minimal stratification through the water column in comparison to previous Austral summer CTD casts. Finally, in December 2013 in the Mertz SW (Figure 1D), we record an increase in salinity and decrease in temperature in the upper water column (~200-400 m): the salinity is 34.56‰ compared to 34.52‰ in 2008, whilst the temperature is -2˚C, somewhat colder than previous years, which cluster around -1.8˚C. Unfortunately, in the Mertz SW the XCTD casts did not reach a sufficient depth to analyse the structure of deeper circulation within the former Mertz Polynya.

**3. Discussion**

Data from the region west of B09B (Commonwealth Bay NW) shows evidence of a shift in water properties following the grounding of B09B in its position during December 2013 (Figure 1B). Prior to the grounding, the water column was stratified, with relatively warm and fresh water overlying a colder, saltier layer. Following the grounding of B09B, the entire water column below 100 dbar has changed, transitioning to become slightly saltier, colder and evidently more well-mixed by 2013 (Figure 1B). Whilst salinity values are not yet as high as those in regions of HSSW production pre-calving, our observations suggest that the Commonwealth Bay NW area maybe becoming an area of deep convection and HSSW formation, in a region where historically no HSSW was formed (Lacarra et al., 2014). Although we cannot discount that this water mass may have been advected from other regions, this interpretation is supported by interpolation of satellite derived sea-ice concentrations, which suggest that sea-ice production in the Commonwealth Bay NW region has been significantly enhanced post-calving of the MGT and grounding of B09B (Figure 2; Tamura et al 2016). These estimates suggest that sea-ice production within the former Mertz Glacier polynya has decreased markedly compared to 2009 levels (Figure 2A; Tamura et al 2016), and has become restricted to an area closer to the coast (Figure 2B; Tamura et al 2016). Contrastingly, sea-ice production in the area of Commonwealth Bay NW, in the lee of the B09B iceberg, is shown to have increased markedly by 2012 (Figures 2B), compared to pre-calving estimates (Figure 2A). Combined, the evidence of enhanced sea-ice production, deep convection and HSSW production in the Commonwealth Bay NW region suggest that an emerging polynya may be developing in the lee of B09B.

Calving of the MGT released a large volume of sea ice from the immediate east of the Mertz Glacier and subsequent melting of the sea ice produced a significant input of fresh water and rapid freshening of the upper ocean post-calving (Shadwick et al., 2013), as seen in Figures 1C and 1D (Green). Our observations hint at a partial recovery of upper ocean salinity by 2013 in the Mertz NE (Figure 1C) and Mertz SW (Figure 1D) regions as of December 2013. Unfortunately, as mentioned, our 2013 XCTD measurements do not extend to sufficient depths to sample the layers below 550m, critical to HSSW production. However,

the apparent reduction in the amount of buoyant fresh water in the upper water column may pre-condition these regions for a resumption or strengthening of HSSW formation in future years, if sufficient formation of sea ice and subsequent brine rejection occurs. Prior to the grounding of B09B in its present position, intrusions of relatively warm modified Circumpolar Deep Water were observed in the Mertz NE region (Figure 1C). Our observations suggest this was not occurring in December 2013, when the upper water column in 2013 was found to be colder (~0.8˚C) and unstratified with respect to temperature.

Our *in situ* observations, in combination with satellite observations, provide valuable insights into ocean dynamics post-MGT calving and grounding of B09B in this region that is critical to HSSW production. Whilst the implications for shifting focus of HSSW on regional AABW formation are unquantified, the changes recorded locally demonstrate that this region is still undergoing marked and dramatic oceanographic changes (Shadwick *et al*., 2013; Clark et al., 2015). To further explore potential future impacts of these changes once the region has re-equilibrated, we use high-resolution ocean modelling to construct pre- and post-calving steady states independently, allowing us to look at the possible implications on regional HSSW production.

**3.1 Exploring the processes driving the new Commonwealth Bay polynya**

To gain an increased understanding of how these regional oceanographic changes triggered by the events that began in 2010 could develop over future years, high-resolution regional ocean model simulations were undertaken to compare two steady states in Commonwealth Bay, both pre- and post-calving of B09B. These were run using a modified Rutgers version of the Regional Ocean Modelling System (ROMS) (Shchepetkin and McWilliams, 2005), with a model setup following Cougnon (2016; see SOM for full model description and set up). The model includes ocean/ice-shelf thermodynamics and frazil ice thermodynamics (Galton-Fenzi et al., 2012), but does not include sea-ice model/ocean coupling. Without a dynamic sea-ice model, the fine-scale polynya activity is resolved by forcing the surface of the model with

monthly heat and salt fluxes from the Tamura *et al*. (2016) dataset, which is based on sea-ice concentration estimated with the Tamura *et al*. (2007) algorithm. This algorithm estimates thin ice thickness using Special Sensor Microwave Imager (SSM/I) observations and the European Centre for Medium-Range Weather Forecast Re-Analysis data (ERA-Interim) (Tamura et al., 2016). In summer the dataset is supplemented

with heat and salt fluxes using monthly climatology from ERA-interim.  The model simulations are forced at the surface with data from the year 2009 (pre-calving; SOM Figure S3) and 2012 (post-calving), providing general information on the ocean circulation for stable ice geometries that includes melt water from the B09B and other fast-ice and icebergs/ice shelves present in the domain. The results from these simulations are not restricted to the year chosen for the forcing, and can be compared with other years of

similar salt and heat flux intensity both pre- and post-calving (see SOM for discussion). The same lateral boundary forcing is used in both pre- and post-calving simulations. Lateral boundary fields, including salinity, horizontal velocities and potential temperature, were relaxed to a climatology calculated from monthly fields estimated from the circulation and climate of the ocean, Phase II synthesis (ECCO2) for the period 1992-2013 (Wunsch, 2009). Each 33 year run includes a spinup phase of 30 years to reach

equilibrium using a repeating loop of the climatology forcing. A climatology of the last 3 years of the run is used for the analysis presented here.

The numerical simulations pre- and post-calving indicate a change in oceanographic conditions in the area of the B09B iceberg, supporting our interpretation of the development of a polynya area in the lee of B09B

post-calving (Figure 3). The modelled ocean circulation for December shows that in the pre-calving simulation, a westward coastal current carries water masses from the Mertz Polynya and Commonwealth Bay regions towards Commonwealth Bay NW (Figures 3A and B), forming a stratified water column with warm and fresh surface water (Figure 3C). The dramatic change in flow from the Mertz Polynya region is shown in more detail in Figure S4. The cold and salty water mass simulated pre-calving at the NW

Commonwealth Bay is advected from the Mertz Polynya and Commonwealth Bay. Modelled water column

stratification is stronger in winter when there is sea-ice production. The model simulates a relatively warm layer at around 150 m depth (-1.18 ˚C) in July pre calving (Figure 3D). From 250 m to the ocean floor there is a cold (-1.92 ˚C) and salty (34.67‰) water mass that originates from the advection of HSSW from the Mertz Polynya and Commonwealth Bay.

Post calving, the coastal current is blocked by the B09B iceberg, associated with a decrease in sea-ice production within the Mertz Polynya (Figure 2); little HSSW is advected into the area of the Commonwealth Bay NW (Figure 3). The model average for December shows a stratified water column in summer, due to the advection from the north of a relatively warm water mass in summer. However, in

winter the water column post-calving in Commonwealth Bay NW is entirely homogeneous in potential temperature (-1.90 ˚C) and salinity (34.54‰), illustrating that under a stable post-calving geometry an active polynya is present, which locally produces HSSW capable of being convected to the sea floor. The seasonality illustrates the potential of a polynya developing in the lee of the B09B iceberg to locally form HSSW dense enough to sink to the sea floor, as inferred from the trends in the summer observations. It

should be noted that our model simulations do not show the current evolution of the impact of the calving, but rather simulate the ocean conditions for two stable ice geometries, before and after the Mertz calving. Thus, whilst our simulations cannot be directly inter-compared to our XCTD data, the trends in both regional circulation and local salinity and temperature provide valuable insights into mechanisms driving circulation changes potentially triggered as a response to the loss of the MGT and the grounding of B09B

off Commonwealth Bay.

**3.2 Implications of data and modelling**

Our XCTD data, in combination with satellite-derived estimates of sea-ice production, indicate that the regional reconfiguration of the Mertz Polynya and Commonwealth Bay and the grounding of iceberg B09B

has had a continuing marked oceanographic impact, suggesting a shift from a equilibrated regime to a

transitional one. High-resolution model simulations suggest that, once re-equilibrated to a new steady state, this may result in a shift in the focus of HSSW production (Figure 1). Data from Commonwealth Bay NW hints at the development of a polynya west of B09B, where today HSSW formation may be taking place outside the previously well-established foci of regional HSSW production in the former Mertz or

Commonwealth Bay polynyas (Lacarra et al., 2014). The effect this change of location will have on regional ocean circulation is currently unquantified, and much of the impact depends on the changes occurring deep in Commonwealth Bay itself under the perennial fast-ice that has formed across the bay triggered by the grounding of B09B (Lacarra et al., 2014; Clark et al., 2015; Cougnon, 2016).

Whilst the observations we present cannot account for seasonal variability (Lacarra et al., 2014), which can only be fully reconciled by the recovery and analysis of the *in situ* CTD arrays deployed in the region, our data and model analysis suggest that water mass characteristics have been affected markedly in the area off Commonwealth Bay and across the former Mertz Polynya. Regardless, our analysis suggest that the grounding of B09B off Commonwealth Bay in 2010 has led to the development of a new polynya to its

leeward side that is capable of producing HSSW outside the Mertz Polynya or the former Commonwealth Bay Polynya (Lacarra et al., 2014), as supported by satellite interpolations of sea ice production (Tamura et al., 2016).

**4. Conclusions**

Before the Mertz Glacier calving event, dense shelf water production from the Adélie shelf supplied 15-25% of the global volume of AABW (Rintoul, 1998). Several studies have documented the decrease in activity of the Mertz and Commonwealth Bay polynyas – and reduction in salinity and density of HSSW – following the calving event (Tamura et al., 2012; Shadwick et al., 2013; Lacarra et al., 2014) and subsequent grounding of B09B. This study captures a unique snapshot of change in key areas of the Adélie

Land continental shelf, and further enhances our understanding of the sensitivity of HSSW formation to

changes in the local icescape. Critically, it illustrates how movement of large icebergs can alter regional ocean circulation and air-sea interaction patterns, producing new polynyas and hence new regions of dense water formation.  While the salinity of HSSW produced in the polynya found in the lee of B09B does not achieve the high values observed in the Mertz and Commonwealth Bay polynyas pre-calving of the MGT, HSSW formed in this new polynya may, in part, compensate for the reduction in dense water production by these now much weaker polynyas. This remarkable 'natural experiment' underscores the sensitivity of HSSW to local changes in the cryosphere and provides insight into the consequences of regional change on ocean circulation.

## 5. Acknowledgements

This work was supported by the Australasian Antarctic Expedition 2013-2014, the Australian Research Council (FL100100195, FT120100004 and DP130104156) and the University of New South Wales. EC is supported by CSIRO and Institute for Marine and Antarctic Studies (University of Tasmania) through the Quantitative Marine Science PhD Program. We would also like to thank Dr Jan Lieser (University of Tasmania) for the sea ice imagery used in Figure 1. Computing resources were provided by both the Tasmanian Partnership for Advanced Computing and the Australian National Computing Infrastructure under grants m68 and gh8. We thank members of the AAE 2013-2014 Captain and crew of the *MV Akademik Shokalskiy* and of the *Aurora Australis,* as well as Australian Antarctic Division Expeditioners.

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

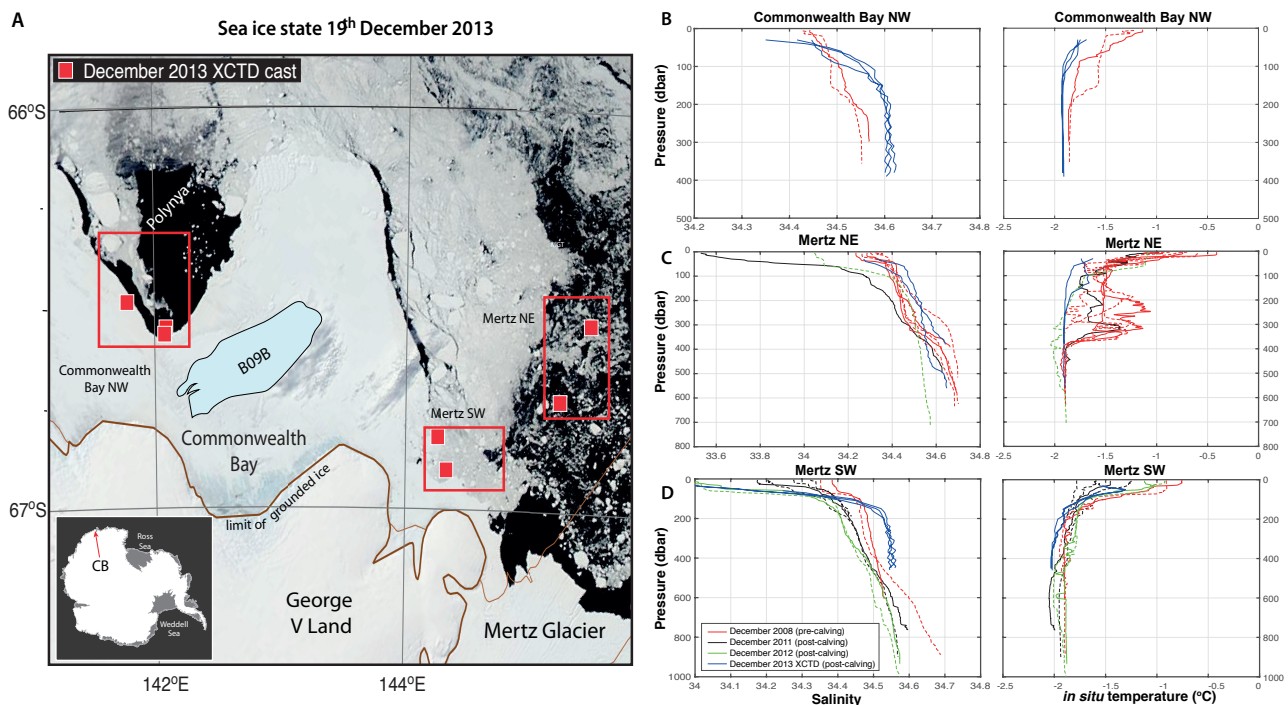

Figure 1. Comparison of new XCTD observations with previous data from the Commonwealth Bay and Mertz Glacier region of Adélie Land, Antarctica. A. Locations of XCTD casts taken in December 2013 on the AAE 2013-2014. The outline of the grounded B09B iceberg is indicated. Base map is visible MODIS image from the 19[th] of December 2013 (credit Dr Jan Lieser: source NASA WORLDVIEW). Inset bottom

10   left: location map (CB Commonwealth Bay). Charts show comparison between salinity and temperature from XCTD casts in 2013 (blue) and CTD profiles from the same month in previous years where data is available for that region (2012:green; 2011:black; 2008, pre-calving: red) from Commonwealth Bay NW (B), NE Mertz (C) and SW Mertz (D). (See SOM Figure S2 for details of specific sites of historic data).

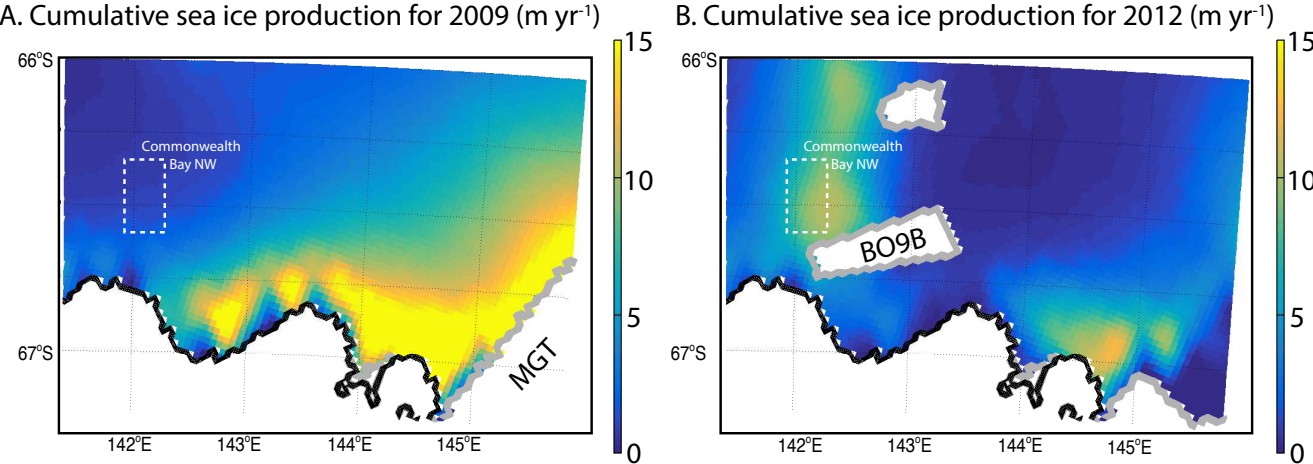

Figure 2. Cumulative sea-ice production estimated from the Special Sensor Microwave Imager (SSM/I) observations for the Mertz and Commonwealth Bay region for pre-2009 (A) and post-2012 (B) MGT calving (after Tamura et al., 2016).

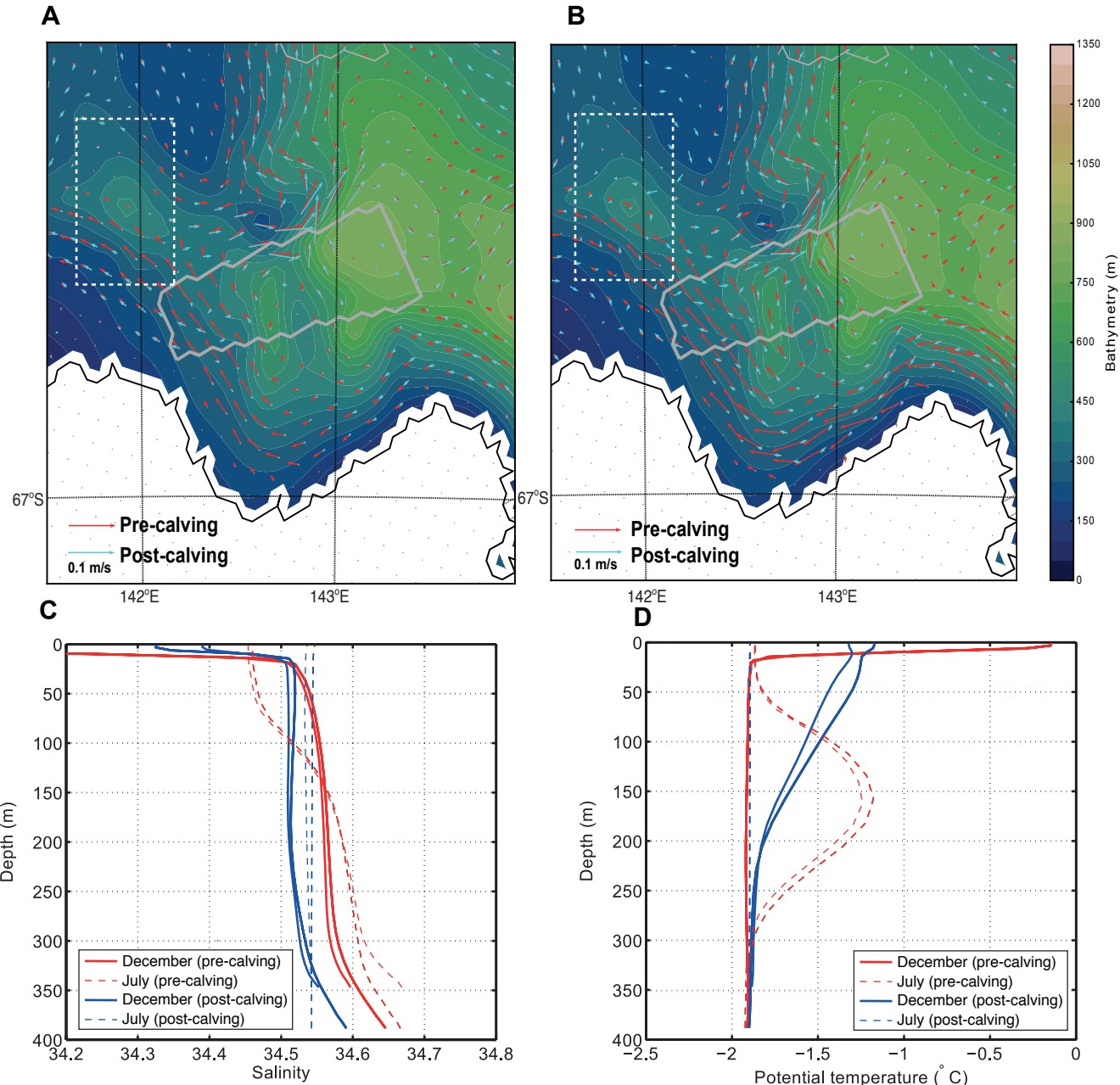

Figure 3. Results of high-resolution model simulations. Upper panels: Simulated bottom current velocity (m/s) from ROMS, averaged on the 5 lowest layers of the model for both pre-calving (red vectors) and post-calving (cyan vectors) geometries near Commonwealth Bay for summer (A. November - December) and winter (B. August – September). The outline of B09B can be seen in light grey, and the location of Commonwealth Bay NW area (white box). Lower panels: modelled salinity (C) and potential temperature (D) from independent simulations with ROMS (n=2), for the Commonwealth Bay NW for 'stable' pre (red) and post calving (blue) geometries, averaged for December (solid lines) and July (dashed lines).