# Peer review of "BRIEF COMMUNICATION: IMPACTS OF A DEVELOPING POLYNYA OFF COMMONWEALTH BAY, EAST ANTARCTICA, TRIGGERED BY GROUNDING OF ICEBERG B09B"

_The Cryosphere, 2016_

## Referee Comment (RC1) · Anonymous Referee #1 · 22 Apr 2016

General comments:

"Evidence of a developing polynya off Commonwealth Bay, East Antarctica, triggered by grounding of iceberg B09B" is a manuscript written with the aim of documenting recent oceanographic observations in a new polynya occurring in the lee of B09B, an iceberg which re-grounded in Commonwealth Bay following the dramatic change in regional icescape in early 2010 associated with the calving of the Mertz Glacier Tongue. These oceanographic observations are supported to some extent by a high-resolution ocean model, of which little description is provided. The text of the manuscript is quite well written, in contrast to the very poor quality of the figures presented.

In summary, it's a manuscript which I would very much enjoy reading if I could a)

read the figures, and b) be convinced that the heat/salt flux used in the pre-calving configuration was appropriate (see details below).

There are some major issues with this manuscript. These are fleshed out in more detail in my specific comments, but I will list those comments that I view as "major" here:

Specific comment 12: the use of 2009 heat/salt flux data for the "pre-calving" model forcing.

Specific comment 13: the description of the model is woefully inadequate.

Specific comment 16: the model results appear to contradict the observations.

Specific comment 17: the quality of both figures is very poor.

Specific comment 21: The shape of iceberg B09B does not appear to be realistic in the model domain.

Specific comments:

The title is somewhat misleading. A polynya is defined as a region of lower sea ice concentration than what would otherwise be expected given the climate of that region. We can easily see that this is a new polynya from satellite imagery/other data. What this paper presents is the oceanographic consequences of this new polynya. As such, I suggest changing the title to reflect this focus.

I have an issue with the recurring reference to the "grounding of B09B". B09B has significantly grounded twice (at least!) since its calving from the Ross Ice Shelf in 1987. As such, reference to the "grounding B09B" is ambiguous. Perhaps change to "recent re-grounding" or similar.

Line 1: "triggered by the impact" – this hasn't been proven. There is an argument that the calving of the MGT was precipitated by the movement of B09B which altered the current configuration, thus calving without an impact. This was outlined in detail by Mayet et al., 2013 (JGR Oceans). This important work on the MGT calving is not cited

once in this manuscript.

P2, L5, other places: The point is made that local changes in the icescape can influence AABW formation, but this ignores the fact that B09B was produced thousands of km away (i.e., not only local, but remote icescape changes can have large impacts).

P2, L11-13: There is more to the existence of the CB polynya than just these factors, i.e., the presence of the MGT, B09B, many smaller grounded icebergs around which fast ice forms, all located upstream of the CB polynya.

P2, L13-15: The text before and after the semicolon is a non sequitur. Also "the sea ice" is ambiguous.

P3, L1-2: As with P2, L5: There is more to the westward flow blocking than just the MGT. See the description provided by Massom et al., 2003 (JGR).

P3, L11-13: This sentence could do with a re-write. Also, mention that while the MGT calving cycle may be cyclic, the re-grounding of B09B in CB is likely to be a "spanner in the works".

P3, L16: For completeness, you need to mention the contribution of Cape Darnley polynya to AABW – see paper by Ohshima et al., 2013 (Nature Geo).

P4, section 2.1: There is absolutely no mention of which month observations were conducted. This seriously adds to confusion in the interpretation of figures.

P4, L24: Tamura et al.'s heat and salt flux data are based on thin ice thickness data, not sea ice concentration data.

P5, L1: I see a major issue with forcing the model using 2009's heat and salt flux data. As shown in Tamura and Williams et al. (2012), 2009 was a strongly anomalous year for sea ice production in the Mertz Glacier polynya. In fact, assuming you didn't use the MODIS fast ice mask (which there is no mention of in this manuscript), 2009 had the highest sea ice production of all years observed. The use of 2009 may have been able to be justified if the pre-calving observations were also conducted in 2009, but they weren't (they were conducted in 2008). While not explicitly stated, it appears that the authors have simply chosen 2009 because it was the year before calving occurred. This appears to be a very poor choice, and has probably influenced the conclusions drawn from the modeling component here. Unless I'm missing something here, I think it would have been much more sensible to force the pre-calving model run using either a more normal year for sea ice production, or a sea ice production climatology based on all of Tamura's years of observation. The choice of 2012 for post-calving seems fine, however. Perhaps the choice of 2009 could be justified by performing a sensitivity analysis? I'm not sure how a sensitivity analysis could be done without rerunning the whole model though.

Section 2.2: The description of the model setup/domain is completely inadequate. What is the resolution? Latitudinal and longitudinal extents? Grid setup? Bathymetry used? What hope does someone have of reproducing your results without this fundamental information? Or are you tasking the text "similar to Cougnon et al., 2013" with providing this information? How similar, exactly? In this case, you need to be more explicit here. And how was the fast ice treated in the model? What was its horizontal extent? Tamura and Williams et al. (2012) highlighted the importance of using accurate fast ice in polynya studies, but the fast ice implementation in the model is not even mentioned here. Was the "dagger" fast ice forming around grounded icebergs to the north of the pre-calving MGT included? This acts to extend the MG polynya (both pre- and post-calving). So many unanswered questions related to the model domain.

P5, L18: "piled up" is a very vague statement, it's possible to be much more exacting. As showed by Massom et al. (2010, JGR Oceans), the very thick fast ice immediately east of the MGT was thermodynamically thickened, and not "piled up" at all. Or are you referring to the largely dynamically-thickened fast ice (which probably was "piled up") east of the pre-2010 grounded position of B09B (see Fraser et al., 2012, Journal of Climate)? I'm not sure which you're referring to, because you state "piled up to the east of the glacier tongue". Do you mean immediately east? Or farther afield?

P5, L20-21: Figure 1D shows very similar salinities in 2008 vs 2011 though.

P6, L7 vs L21. The comment is made in L7 that the post-grounding water column is saltier than pre-calving, based on observations. However, in L21 you say that the post-grounding water column is fresher, based on model results. This is also manifested in Fig 1A vs Fig 2C. No mention of this discrepancy is made in the text of this paper. It seems like a major failure of the model to reproduce the observations. Could you make a comment about this?

P7, L22: The blocking of the coastal current is a major result of the model, yet its importance is not emphasized anywhere in the discussion. Here might be a good place to include it.

P8, L2: How does the sea ice production compare between your pre and post-calving years?

P8, L10: Is HSSW formed in this region able to go on to form AABW? A comment on the bathymetry in the region of the new polynya would be appropriate here.

Figure 1 is very poorly presented. There are numerous typos (Decmebr and Tounge). The font size varies wildly across the figure, much of the text is illegible. Both inset maps for Fig 1A are almost useless. The lower left one really suffers from not having a coastline drawn. Fig 1A needs much more annotation. What is continent? What is fast ice" What is pack ice? How does the date of acquisition of this image relate to the time of field observations? Fig 1A should be zoomed out a little to provide more context – we can't even see the "original" B09B grounding location or the full extent of the tongue. There is absolutely no representation of the icescape pre-calving! The caption is confusing in the way that it references the sub-figures (and doesn't even mention sub-figures E, F or G). The figure refers to both B9B and B09B. The color "blue" is given a capital letter in the caption for some reason (and "red" doesn't even rate a mention).

Figure 2 is very poorly presented. Summer and winter figures seem randomly placed. Wouldn't it be a good idea to arrange all "winter" figures on the left, and all "summer" figures on the right? And why does "Nov-Dec" appear before "Aug-Sep"? It's chronologically backward. Fig 2B has no label on the legend. This figure is completely illegible in print, and only slightly better online. There's a fundamental problem with the presentation of Figures 2A and 2B: since the pre-calving vectors are directly over-plotted on the post-calving vectors, and there's no translucency, then it's impossible to assess if the underlying vector if the overlying vector completely obscures it. It's a terri-bly unreadable way to present two vector fields. At the very least, one series of vectors should be offset slightly. Possibly most importantly, the outline of B09B appears to bear little resemblance to the shape of that in Fig 1. Why is the eastern end of B09B not tapered in the model domain? B09B is referred to as both "B09B" and "B09b" in the caption. Finally, the caption could use some revisions, English-wise – some strange sentences as well as some parenthesis nastiness.

Figure S1 adds very little to this manuscript. It would be sufficient to say that the xctd matches the microcat values very closely (possibly give an RMS difference, or similar measure of agreement).

Technical corrections:

Be consistent with capitalization of T in Mertz Glacier Tongue.

Strange bracket situation on P2, L21. Strange space situation there too.

P3, L18: Replace "are" with "is".

P4, L22: Double reference to Cougnon et al., 2013.

Be consistent with "fast-ice" vs "fast ice".

P3, L5: check capitalization for the C, C and O parts of ECCO.

P6, L1: Colder than. . .?

P6, L8: Replace "occurred" with "now occurs".

P6, L18: Location of Adelie Depression needs to be shown in a figure.

P6, L5: I don't like the use of "present position" for B09B. What if it moves? Better to tie the description to a year or epoch.

P8, L15: Counter parts should be counterparts.

Again, I'd like to reinforce that I think this kind of experiment is very interesting, and would like to see a revised version which can convince me that the heat/salt flux forcing (pre-calving) is appropriate (a reference to specific comment number 12). I hope the authors can take the time to convince me, or to run the model again with more appropriate heat/salt flux forcing.

---

## Referee Comment (RC2) · Anonymous Referee #2 · 28 Apr 2016

General Comments: This paper discusses a newly-formed DSW (Dense Shelf Water) formation region in the western side of large iceberg B9B, based on direct oceanographic observations and a high-resolution regional numerical ocean modeling. The Mertz Glacier Tongue (MGT) and B9B largely changed the position in January-February 2010. This region had been known as one of the most active formation regions of sea ice and DSW around Antarctica. Thus, although it is a very regional change, the direct ocean observations after the MGT calving event are valuable for better understanding the changing Southern Ocean and Antarctic Cryosphere. Ocean modeling would be useful for interpreting the sparse sampling of observations in the space and time if the model realistically could produce ocean circulation and water

masses in the focal region (i.e., the Adelie Depression for this study).

In my reading, unfortunately, the results from observation and modeling did not show the DSW formation in the lee side of B9B. I have seven main concerns, I'm happy if these comments are helpful for revising your manuscript.

[1] Although this paper speculated the local DSW formation in the lee side of B9B from the T-S profiles (Fig.1 B and E), T-S profiles in the Mertz SW region (Fig.1 D and G) have also a similar structure. There is a possibility that DSW is advected from the east.

[2] Showing a summer image in Fig. 1A is misleading. Polynyas in winter and spring have different roles. While winter polynya plays a role in high sea ice and DSW productions, spring polynya is a sea ice melting area. It seems to me that showing winter sea ice concentration or sea ice production is a direct way to indicate an active formation region of sea ice and DSW.

[3] The ocean model failed to reproduce the ocean properties. The observation (Fig. 1 B) shows an increase in summer salinity, but model does not. The temperature profiles are also different between the two.

[4] There are no pronounced differences in ocean velocity. In first place, how can you speculate the polynya activity from ocean velocity? I expect that a (bottom) salinity field is more suitable to show the activity before and after the relocation of B9b.

[5] Figure 1 should be revised. It is too small to see. Larger area which covers the Adelie Depression and the MGT is preferable. Please add bottom contour, grounding line, and ice front line to easily understand the regional configuration. I expect that active sea ice production near the B9B is on the Adelie Bank, not the Adelie Depression. If so, it seems to be difficult for the local water to be dense enough and to be exported from the Adelie Sill (where is the main pathway of DSW formed in the Adelie Depression).

[6] More detail of the model configuration is required in section 2.2. Model description in

the present form was insufficient for me to understand the model setting. My concerns about the model setting are ... (1) Horizontal/vertical resolution (2) Initial condition (the same condition for the two experiment?) (3) Why did you select 2009 and 2012 forcing? How well do the two years represent the conditions before and after the MGT calving/B9B relocation? It is helpful for readers to show maps of surface salinity flux (sea ice production) used for the model. (4) How long was the model integrated for each experiment?

[7] There are several sentences throughout the manuscript to speculate the impact on AABW. I don't think that emphasizing the connection to AABW at many place is important because this paper examined only the polynya near one large iceberg without showing the relative importance in the total DSW and AABW production. Some of them should be removed.

Minor comments

[Introduction] It is helpful for readers to describe the history of the B9B movement (at least after the MGT calving event).

[P5, L20-21] I can't see the salinity recovery in Fig. 1D.

[P5, L21-24] Without discussing the interannual variability, it is impossible to speculate the preconditioning of coastal ocean by the icescape change.

[P7, L13, "The effect this change ..."] I think you can check the ocean circulation change in the model.

[Figure 1] What do the dashed and solid lines indicate? I suggest that figures for the place/icescape and T-S profiles be shown separately to enlarge each figure.

[Figure 2] Background color for depth (only green and blue) does not make sense.

[Figure]

---

## Author Response (AR1)

**BRIEF COMMUNICATION: IMPACTS OF A DEVELOPING POLYNYA OFF COMMONWEALTH BAY, EAST ANTARCTICA, TRIGGERED BY GROUNDING OF ICEBERG B09B**

5  Christopher J. Fogwill[1*], Erik van Sebille[2], Eva A. Cougnon[3,4,5], Chris S.M. Turney[1], Steve R. Rintoul[3,4,5], Benjamin K. Galton-Fenzi[4,6], Graeme F. Clark[1], E.M. Marzinelli[1], Eleanor B. Rainsley[7], Lionel Carter[8].

[1] Climate Change Research Centre, School of Biological, Earth and Environmental Sciences, UNSW Sydney, Australia

[2] Grantham Institute & Department of Physics, Imperial College London, United Kingdom

10  [3] Institute for Marine and Antarctic Studies, University of Tasmania, Private Bag 129, Hobart, Tasmania 7001, Australia.

[4] Antarctic Climate & Ecosystems Cooperative Research Centre, University of Tasmania, Private Bag 80, Hobart, Tasmania 7001

[5] Commonwealth Scientific and Industrial Research Organisation, Ocean and Atmospheric Research, Hobart, Australia.

[6] Australian Antarctic Division, Kingston, Tasmania

15  [7] Wollongong Isotope Geochronology Laboratory, School of Earth and Environmental Sciences, University of Wollongong, Wollongong, Australia.

[8] Antarctic Research Centre, Victoria University of Wellington, New Zealand

*Correspondence to c.fogwill@unsw.edu.au

**Response to reviewers**

We thank each of the reviewers for their detailed reviews, and in light of their comments and suggestions have updated our manuscript as outlined in the following paragraphs.

The reviewers raise two key issues. Firstly, they request more details of the model, and question suitability of 2009 climatology in driving the pre-calving simulation. Secondly, they raise questions over the apparent contradiction between our observations and model simulations. In our resubmission we address these points directly, with an expanded model description and climatological analysis (within the main text, and the supplementary information), and perform further analysis of the trends in circulation and sea ice production off Commonwealth Bay in the model simulations that support our assertion that a polynya has developed in the lee of iceberg B09B. In addition, we have reformatted the figures within the main body of the text in line with the reviewer's comments, and include additional figures in the supplementary files to address the reviewer's specific comments as outlined in the paragraphs below. Again, we wish to thanks the reviewers for their detailed comments that have substantially strengthened this Brief Communication in the *Cryosphere*.

**Reviewer One**

1 The title is somewhat misleading. A polynya is defined as a region of lower sea ice concentration than what would otherwise be expected given the climate of that region. We can easily see that this is a new polynya from satellite imagery/other data. What this paper presents is the oceanographic consequences of this new polynya. As such, I suggest changing the title to reflect this focus.

1. We understand the reviewers point here, and as such have reworded the title to, "IMPACTS OF A DEVELOPING POLYNYA OFF COMMONWEALTH BAY, EAST ANTARCTICA, TRIGGERED BY GROUNDING OF ICEBERG B09B"

2 I have an issue with the recurring reference to the "grounding of B09B". B09B has significantly grounded twice (at least!) since its calving from the Ross Ice Shelf in 1987. As such, reference to the "grounding B09B" is ambiguous. Perhaps change to "recent re-grounding" or similar.

2. We recognise the reviewer's point regarding the long-history and multiple groundings of B090B since its calving in 1987, but here we are discussing the instance of grounding in Commonwealth Bay post Mertz Glacier Tongue calving in 2010, as outlined in the title, abstract and Page 2, Line 15 of the manuscript. There are many other manuscripts (many of which we include within the 20 references available) that detail the history and events prior to and during the Mertz Glacier calving in 2010, but this is not the purpose of this Brief Communication for the *Cryosphere*, which aims to understand the observations from 2013 from Commonwealth Bay through our model analysis.

3 Line 1: "triggered by the impact" – this hasn't been proven. There is an argumen that the calving of the MGT was precipitated by the movement of B09B which altered the current configuration, thus calving without an impact. This was outlined in detail by Mayet et al., 2013 (JGR Oceans). This important work on the MGT calving is not cited

3. We acknowledge this point, and have changed the wording in Line 1 to, '…precipitated by the movement…". We agree that Mayet et al., 2013 provides an important discussion of the hydrological events during the MGT calving event, but are unfortunately limited as to the number of references we can include.

4 P2, L5, other places: The point is made that local changes in the icescape can influence AABW formation, but this ignores the fact that B09B was produced thousands of km away (i.e., not only local, but remote icescape changes can have large impacts).

4. We remove the word 'local' from line 6, page 2, and 'regional' from line 8, page 3, to acknowledge that changes in the icescape away from Adelie Land may also have an affect in this region.

5 P2, L11-13: There is more to the existence of the CB polynya than just these factors, i.e., the presence of the MGT, B09B, many smaller grounded icebergs around which fast ice forms, all located upstream of the CB polynya.

5. We thank the reviewer for this comment – we are here talking about the historically present Mertz polynya as opposed to the newly created one in the lee of B09B off Commonwealth Bay, and have changed the wording of Page 2, Lines 13 and 14 to clarify this.

6 P2, L13-15: The text before and after the semicolon is a non sequitur. Also "the sea ice" is ambiguous.

6.   We have changed the wording of this sentence to make its meaning more obvious.

> 7 P3, L1-2: As with P2, L5: There is more to the westward flow blocking than just the MGT. See the description provided by Massom et al., 2003 (JGR).

7.   We have changed 'blocked' to 'reduced' to reflect this point.

> 8 P3, L11-13: This sentence could do with a re-write. Also, mention that while the MGT calving cycle may be cyclic, the re-grounding of B09B in CB is likely to be a "spanner in the works".

8.   We thank the reviewer for this comment, and have restructured the sentence accordingly, and highlighted that we

are talking about the impacts of MGT calving on AABW formation being cyclical, as opposed to the appearance of

'megabergs' such as B09B.

> 9 P3, L16: For completeness, you need to mention the contribution of Cape Darnley polynya to AABW – see paper by Ohshima et al., 2013 (Nature Geo).

10   9.   We thank the reviewer for suggesting this and in our updated manuscript we add the AABW sites at Cape Darnley

and off Vincennes Bay.

> 10 P4, section 2.1: There is absolutely no mention of which month observations were conducted. This seriously adds to confusion in the interpretation of figures.

10.   We thank the reviewer for spotting this omission, and have added the month (December 2013) accordingly.

> 11 P4, L24: Tamura et al.'s heat and salt flux data are based on thin ice thickness data, not sea ice concentration data.

15   11.   We acknowledge this, and have updated this section of the manuscript and include further details both in the text

and in the supplement.

> 12 P5, L1: I see a major issue with forcing the model using 2009's heat and salt flux data. As shown in Tamura and Williams et al. (2012), 2009 was a strongly anomalous year for sea ice production in the Mertz Glacier polynya. In fact, assuming you didn't use the MODIS fast ice mask (which there is no mention of in this manuscript), 2009 had the highest sea ice production of all years observed. The use of 2009 may

have been able to be justified if the pre-calving observations were also conducted in 2009, but they weren't (they were conducted in 2008). While not explicitly stated, it appears that the authors have simply chosen 2009 because it was the year before calving occurred. This appears to be a very poor choice, and has probably influenced the conclusions drawn from the modeling component here. Unless I'm missing something here, I think it would have been much more sensible to force the pre-calving model run using either a more normal year for sea ice production, or a sea ice production climatology based on all of Tamura's years of observation. The choice of 2012 for post-calving seems fine, however. Perhaps the choice of 2009 could be justified by performing a sensitivity analysis? I'm not sure how a sensitivity analysis could be done without rerunning the whole model though.

12. The choice of the year 2009 for the PRE simulation forcing was made after analysing the monthly heat and salt fluxes averaged over the Mertz Glacier Polynya (MGP) area for the period 1992 to 2013 from Tamura et al., (2016; Figure 1). The period from 2007 to 2009 was identified as a constant sustained period with a winter average (May to September inclusive) of about -164 W m$^{-2}$, while the average over the pre-calving period (1992-2009) is of -159±17 W m$^{-2}$. Similarly, the salt fluxes averaged for 2007-2009 is of about 0.82 kg m$^{-2}$, while the averaged for 1992 to 2009 is of 0.82±0.1 0.82 kg m$^{-2}$. As a result, 2007 to 2009 is considered as being a representative period for the pre-calving MGP region. Ultimately 2009, the year closest to the calving, was chosen as the main purpose of the simulations in this study is to illustrate the general ocean conditions related to a stable ice geometry pre- and post-calving. Also, a single year forcing was preferable to a pre-calving climatology, when compared to a single year forcing for the post-calving simulation, that is restricted to one year due to data availability. In the post-calving scenario, 2012 was chosen in consideration of the fast ice and its representation of permanent features between 2010 and 2012 (A. Fraser personal communication). In summary, the results from these simulations are not restricted to the year chosen for the forcing, they can be compared with other years of similar salt and heat flux intensity as seen from Figure S3. We have added this detail in full to the supplementary information, due to length

restrictions of the main body.

[Figure]

**Figure 1. Monthly surface heat (a) and salt (b) fluxes averaged over the Mertz Glacier Polynya (MGP) from Tamura et al., (2016) data set, with winter time averages (May to September inclusive) shown with crosses (Cougnon 2016).**

13 Section 2.2: The description of the model setup/domain is completely inadequate. What is the resolution? Latitudinal and longitudinal extents? Grid setup? Bathymetry used? What hope does someone have of reproducing your results without this fundamental information? Or are you tasking the text "similar to Cougnon et al., 2013" with providing this information? How similar, exactly? In this case, you need to be more explicit here. And how was the fast ice treated in the model? What was its horizontal extent? Tamura and Williams et al. (2012) highlighted the importance of using accurate fast ice in polynya studies, but the fast ice implementation in the model is not even mentioned here. Was the "dagger" fast ice forming around grounded icebergs to the north of the pre-calving MGT included? This acts to extend the MG polynya (both pre- and post-calving). So many unanswered questions related to the model domain.

13. We have now fully outlined the model description in the supplementary material to cover all these points as outlined below:

*The model used here is based on the Rutgers version of the Regional Ocean Modeling System (ROMS) (Shchepetkin and*
5 *McWilliams, 2005) that includes ocean/ice shelf and frazil ice thermo-dynamics (Galton-Fenzi et al., 2012; Dinniman et al., 2003). The horizontal and vertical grid is the same than presented in Cougnon (2016). Without a dynamic sea ice model, the fine-scale polynya activity is resolved by forcing the surface of the model with monthly heat and salt from Tamura et al. (2016) data set that is based on sea ice concentration estimated with the Tamura et al. (2007) algorithm. This algorithm estimates thin ice thickness using Special Sensor Microwave Imager (SSM/I) observations and the*
10 *European Centre for Medium-Range Weather Forecast Re-Analysis data (ERA-Interim). Water masses formed on the continental shelf in the model are controlled by the variability of the air/sea forcing as well as by the glacial melt water released from the local ice shelves. The model has been set up to compare the ocean and basal ice shelf melting changes post-calving compared with other years of similar heat and salt fuxes intensity within the MGP region. The year 2009 and 2012 are chosen for the pre- and the post-calving air/sea forcing simulations respectively, after analysing the*
15 *monthly heat and salt fluxes averaged over the Mertz polynyna area for the period 1992 to 2013 (Figure S3). The year 2009 is representative to an average to strong sea ice production year in terms of heat and salt fluxes and 2012 was chosen in consideration of the fast ice and its representation of permanent features between 2010 and 2012 (A. Fraser personal communication). Fast ice is parameterised as in Cougnon et al. (2013) and Cougnon (2016), using an updated version of Fraser et al. (2012). Lateral boundary fields, including salinity, potential temperature and horizontal*
20 *velocity, were relaxed to a climatology calculated from the monthly fields from the Estimating Circulation and Climate of the Ocean, Phase II synthesis (ECCO2) for the period 1992-2013 (Wunsch et al., 2009). It is important to note that salinity values used in the model are on the Practical Salinity Scale (PSS78) and are dimensionless. The total run time of the model simulation was 33 years for each simulation. This 33 year run includes a spinup phase of 30 years to reach*

*equilibrium using a repeating loop of the climatology forcing. A climatology of the last 3 years of the run are used for the analyses.*

14 P5, L18: "piled up" is a very vague statement, it's possible to be much more exacting. As showed by Massom et al. (2010, JGR Oceans), the very thick fast ice immediately east of the MGT was thermodynamically thickened, and not "piled up" at all. Or are you referring to the largely dynamically-thickened fast ice (which probably was "piled up") east of the pre-2010 grounded position of B09B (see Fraser et al., 2012, Journal of Climate)? I'm not sure which you're referring to, because you state "piled up"

14. We understand the point the reviewer makes here, and have changed the wording accordingly. We are discussing the geographical source of the sea ice, as opposed to the mechanism by which it built up.

15 P5, L20-21: Figure 1D shows very similar salinities in 2008 vs 2011 though.

15. We thank the reviewer for this comment, but respond that the profiles of the salinities in 2008 and 2011 differ markedly. For example, at 200 m, in 2008 the salinity was 35.48, whilst in 2011 it was 34.43, a significant difference.

16 P6, L7 vs L21. The comment is made in L7 that the post-grounding water column is saltier than pre-calving, based on observations. However, in L21 you say that the post-grounding water column is fresher, based on model results. This is also manifested in Fig 1A vs Fig 2C. No mention of this discrepancy is made in the text of this paper. It seems like a major failure of the model to reproduce the observations. Could you make a comment about this?

16. We acknowledge that this difference has not been fully explored in our original manuscript therefore in our updated manuscript we add the following text to section 3.2:

*The numerical simulations pre- and post-calving indicate a change in oceanographic conditions in the area of the B09B iceberg, demonstrating the development of a polynya area in the lee of B09B post-calving. The modelled sea-ice production (Tamura et al 2016) within the Mertz Glacier polynya decreases and is restricted to an area closer to the coast. On the other hand, sea-ice production in the lee of the B09B iceberg post-calving is shown to increase markedly (Figures 2A and B).*

*The modelled ocean circulation for December shows that pre-calving, a westward coastal current carried water masses from the Mertz polynya and Commonwealth Bay areas towards the Commonwealth Bay NW XCTD positions (red squares on Figures 2A and B), forming a stratified water column with warm and fresh surface water (Figure 2C). The cold and salty water mass simulated pre-calving at the NW Commonwealth Bay XCTD positions is advected from the Mertz polynya and Commonwealth Bay post-calving. Modelled water column stratification is stronger in winter when there is sea-ice production. The model simulates a relatively warm layer at around 150 m depth (-1.18 ˚C) in July pre calving (Figure 2D). From 250 m to the ocean floor there is a cold (-1.92 ˚C) and salty (34.67) water mass that originates from the advection of HSSW from the Mertz polynya and Commonwealth Bay.*

*Post calving, the coastal current is blocked by the B09B iceberg, associated with a decrease in sea ice production within the Mertz polynya; little HSSW is advected into the area of the Commonwealth Bay NW XCTDs. The model average for December shows a stratified water column in summer, due to the advection from the north of a relatively warm water mass in summer. However, the water column post calving at the Commonwealth Bay NW XCTDs is entirely homogeneous in potential temperature (-1.90 ˚C) and salinity (34.54), illustrating an active polynya that locally produces HSSW capable of being convected to the sea floor in winter. The model does not simulate an increase in salinity post-calving, but the seasonality illustrates the potential of a polynya developing in the lee of the B09B iceberg to locally form HSSW dense enough to sink to the sea floor, as inferred from the trends in the summer observations. It should be noted that our model simulations do not show the current evolution of the impact of the calving, but rather simulate the ocean conditions for two stable ice geometries, before and after the Mertz calving, thus can not be directly inter-compared to our XCTD data. However, the trends indicated from our regional model simulations provide valuable insights into mechanisms driving the circulation changes triggered as a response to the grounding of B09B off Commonwealth Bay.*

17 P7, L22: The blocking of the coastal current is a major result of the model, yet its importance is not emphasized anywhere in the discussion. Here might be a good place to include it.

17.  We thank the reviewer for highlighting this important model result, and on Page 6, Line 18, comment upon this and

also in the discussion, Page 7, Line 11.

> 18 P8, L2: How does the sea ice production compare between your pre and post-calving years?

18.  We add further information of this to Page 8, Lines 14-16, supported by new panels on Figure 2 to show sea-ice

production pre and post calving.

> 19 P8, L10: Is HSSW formed in this region able to go on to form AABW? A comment on the bathymetry in the region of the new polynya would be appropriate here.

19.  We have added discussion on the contribution of HSSW form this region to AABW formation to Page7, Lines 10 to

11, and also in the Supplementary Information.

> 20 Figure 1 is very poorly presented. There are numerous typos (Decmebr and Tounge). The font size varies wildly across the figure, much of the text is illegible. Both inset maps for Fig 1A are almost useless. The lower left one really suffers from not having a coastline drawn. Fig 1A needs much more annotation. What is continent? What is fast ice" What is pack ice? How does the date of acquisition of this image relate to the time of field observations? Fig 1A should be zoomed out a little to provide more context – we can't even see the "original" B09B grounding location or the full extent of the tongue. There is absolutely no representation of the icescape pre-calving! The caption is confusing in the way that it references the sub-figures (and doesn't even mention sub-figures E, F or G). The figure refers to both B9B and B09B. The color "blue" is given a capital letter in the caption for some reason (and "red" doesn't even

20.  Figure 1 has been updated and reformatted in line with the reviewer's suggestions.

[Figure]

21 Figure 2 is very poorly presented. Summer and winter figures seem randomly placed. Wouldn't it be a good idea to arrange all "winter" figures on the left, and all "summer" figures on the right? And why does "Nov-Dec" appear before "Aug-Sep"? It's chronologically backward. Fig 2B has no label on the legend. This figure is completely illegible in print, and only slightly better online. There's a fundamental problem with the presentation of Figures 2A and 2B: since the pre-calving vectors are directly over-plotted on the post-calving vectors, and there's no translucency, then it's impossible to assess if the underlying vector if the overlying vector completely obscures it. It's a terri-bly unreadable way to present two vector fields. At the very least, one series of vectors should be offset slightly. Possibly most importantly, the outline of B09B appears to bear little resemblance to the shape of that in Fig 1. Why is the eastern end of B09B not tapered in the model domain? B09B is referred to as both "B09B" and "B09b" in the caption. Finally, the caption could use some revisions, English-wise – some strange sentences as well as some parenthesis nastiness.

21. Figure 2 has been updated in our resubmission, with the panels now showing the simulated changes in vertically integrated velocity and cumulative sea ice production, together with the changes in salinity and temperatures in the model domain. The 'shape' of B09B is realistic in the scale of the model domain, but we acknowledge does not perfectly capture the shape of B09B in the Landsat image.

[Figure]

22 Figure S1 adds very little to this manuscript. It would be sufficient to say that the xctd matches the microcat values very closely (possibly give an RMS difference, or similar measure of agreement).

22. The supplementary figures now include Figure S1, S2 and S3, which add important information. Figure S1 is kept

in the supplement for completeness.

Furthermore, we have addressed each of the technical corrections highlighted by Reviewer One.

**Reviewer Two**

[1] Although this paper speculated the local DSW formation in the lee side of B9B from the T-S profiles (Fig.1 B and E), T-S profiles in the Mertz SW region (Fig.1 D and G) have also a similar structure. There is a possibility that DSW is advected from the east.

1.  We appreciate the reviewer's point in regards to advection of water masses from the East, but our modelling argues against this hypothesis. This can be seen in Figure 2 and is discussed in the text with our vertically integrated velocity profile suggesting little advection from the East due to the blocking effect of B09B.

[2] Showing a summer image in Fig. 1A is misleading. Polynyas in winter and spring have different roles. While winter polynya plays a role in high sea ice and DSW productions, spring polynya is a sea ice melting area. It seems to me that showing winter sea ice concentration or sea ice production is a direct way to indicate an active formation region of sea ice and DSW.

2.  Figure 1 is provided for context, not necessarily to show the polynya activity per se.

[3] The ocean model failed to reproduce the ocean properties. The observation (Fig. 1 B) shows an increase in summer salinity, but model does not. The temperature profiles are also different between the two.

3.  We thank the reviewer for this comment, and have addressed this point as per point 16 of Reviewer One's comments.

[4] There are no pronounced differences in ocean velocity. In first place, how can you speculate the polynya activity from ocean velocity? I expect that a (bottom) salinity field is more suitable to show the activity before and after the relocation of B9b.

4.  We now present the vertically integrated horizontal velocity together with the changes predicted in sea ice production which demonstrate the changes in ocean circulation together with the changes in sea ice production post grounding of B09B.

[5] Figure 1 should be revised. It is too small to see. Larger area which covers the Adelie Depression and the MGT is preferable. Please add bottom contour, grounding line, and ice front line to easily understand the regional configuration. I expect that active sea ice production near the B9B is on the Adelie Bank, not the Adelie Depression. If so, it seems to be difficult for the local water to be dense enough and to be exported from the Adelie Sill (where is the main pathway of DSW formed in the Adelie Depression).

5. We address this in point 16 in our response to Reviewer One

[6] More detail of the model configuration is required in section 2.2. Model description in

6. We have added further detail of the model setup to the text supported in detail in the supplementary information, as

5       outlined in point 13 to Reviewer One.

[7] There are several sentences throughout the manuscript to speculate the impact on AABW. I don't think that emphasizing the connection to AABW at many place is important because this paper examined only the polynya near one large iceberg without showing the relative importance in the total DSW and AABW production. Some of them should be removed.

7. We recognise the reviewers point here and have addressed this throughout our revised manuscript.

We have addressed each of the minor comments highlighted by Reviewer One in the text of our updated manuscript.

10   We thank the reviewers for their input and detailed comments.

Dr Chris Fogwill on behalf of the co-authors.

[revised manuscript text omitted]

The casts

| Page 5: [2] Deleted | Christopher Fogwill | 7/06/2016 2:44 PM |

).

**3.2 Comparison with past data**

| Page 10: [3] Deleted | Christopher Fogwill | 7/06/2016 2:44 PM |

Galton-Fenzi, B. K., Hunter, J. R., Coleman, R., Marsland, S. J., and Warner, R. C.: Modeling the basal melting and marine ice accretion of the Amery Ice Shelf, Journal of Geophysical Research: Oceans, 117, C09031, 10.1029/2012JC008214, 2012.

Harris, P. T., Brancolini, G., Armand, L., Busetti, M., Beaman, R. J., Giorgetti, G., Presti, M., and Trincardi, F.: Continental shelf drift deposit indicates non-steady state Antarctic bottom water production in the Holocene, Marine Geology, 179, 1-8, http://dx.doi.org/10.1016/S0025-3227(01)00183-9, 2001.

| Page 14: [4] Deleted | Christopher Fogwill | 7/06/2016 2:44 PM |

[Figure]

[Figure]

Page 15: [5] Deleted                    Christopher Fogwill                    7/06/2016 2:44 PM

[Figure]

Page 15: [5] Deleted                                          Christopher Fogwill                                      7/06/2016 2:44 PM

[Figure]

[Figure]

[Figure]

Page 15: [5] Deleted          Christopher Fogwill          7/06/2016 2:44 PM

[Figure]

Page 15: [5] Deleted                    Christopher Fogwill                    7/06/2016 2:44 PM

[Figure]

[Figure]

Page 15: [5] Deleted       Christopher Fogwill       7/06/2016 2:44 PM

[Figure]

[Figure]

Page 15: [5] Deleted        Christopher Fogwill        7/06/2016 2:44 PM

[Figure]

Page 15: [5] Deleted         Christopher Fogwill         7/06/2016 2:44 PM

[Figure]

[Figure]

Page 15: [5] Deleted         Christopher Fogwill         7/06/2016 2:44 PM

[Figure]

[Figure]

Page 15: [5] Deleted | Christopher Fogwill | 7/06/2016 2:44 PM

---

## Author Response (AR2)

**Dr Chris Fogwill**
**ARC Future Fellow**
**Climate Change Research Centre**
University of New South Wales
Sydney, NSW, Australia, 2052
Tel: +61 2 9385 7065 Fax: +61 2 9385 8969
Email: C.Fogwill@unsw.edu.au

Dear Dr Matsuoka,

Thank you so much for your further Editorial review of manuscript tc-2016-19 for *The Cryosphere*. Please find attached a revised resubmission of our manuscript entitled 'IMPACTS OF A DEVELOPING POLYNYA OFF COMMONWEALTH BAY, EAST ANTARCTICA, TRIGGERED BY GROUNDING OF ICEBERG B09B' which remains formatted as a Brief Communication for *The Cryosphere.* In light of your and both reviewers comments we have substantiality revised the results and discussion sections of the manuscript, to focus principally on the observational data as suggested, disentangling the observational data from the model projections. We firmly believe that the modelling outputs (based on multidecadal runs) add greatly to the single season observations, and are therefore critical to understand the mechanisms driving contemporary circulation changes in this important region; a region which is still undergoing substantial change.

Our previous submission did not clearly differentiate current observations of the region's transitional state from modelled projections of a future re-equilibriated steady-state, and this has obviously led to confusion. We hope that our updated manuscript clarifies the issues raised. In our resubmission we attempt to highlight that the aim of the modelling study was not simply to '*reproduce ocean properties*' (Reviewer Two), either before or after the Mertz Glacier calving event or the grounding of B09B in Commonwealth Bay, but rather to understand the mechanisms of change and potential future impacts across the region. Each of our model simulations represents the circulation after 30 years under the respective geometries, and therefore is unlikely to '*perfectly*' (Reviewer Two) simulate this complex system, which, importantly, remains in flux after the dramatic events of 2010.

It is critical to clarify the complementary aims of the dual approach taken in this study: the observations provide evidence of a developing polynya whilst the multidecadal model runs

provide mechanistic insights into the processes involved. To communicate this more clearly, we have substantially reformatted the manuscript to disentangle our observational data from the modelled scenarios. We have also added a further figure (Figure 2), which depicts the impacts on sea-ice production based upon the Tamaura et al., (2016) sea ice reconstructions for 2009 and 2012, which support our interpretation of a developing polynya in the lee of B09B. As asserted in our original resubmission, *'our model simulations do not show the current evolution of the impact of the calving, but rather simulate the ocean conditions for two stable ice geometries, before and after the Mertz calving'*; however, we argue that our modelling provides insights into the potential circulation changes (defined from simulated current velocity for the bottom 5 layers of the model), and that the simulated salinity and temperatures are useful measures with which to explore the potential future impacts of the event, as well as the mechanisms that continue to unfold.

Therefore, combined, the current observational data of a dynamic system together with modeled future projections enhance our understanding of the sensitivity of HSSW and AABW formation to changes in the local icescape. Our results show how movement of large icebergs such as B09B can alter regional ocean circulation and air-sea interaction patterns, producing new regions of dense water formation. Critically, we feel these observations are well suited to the format of a *Cryosphere* 'Brief Communication'.

In response to your review we respond to the following key points:

1. *If modelled velocity data is used to support the HSSW advection, the modelled velocity should be presented at the depths where HSSW is expected:*

We have updated the revised Figure 3 to demonstrate this, by presenting the velocity changes in the **lowest 5 layers** of the model, the result of which support our original interpretation. The blocking effects of B09B on water masses from the Mertz region can be clearly seen.

[Figure]

2.  *'The author also claimed that there is an active polynya in Commonwealth Bay NW and generate HSSW that is convected to the sea floor (P7L11). However, it is unclear for me whether this water mass with the uniform salinity is locally made or advected from somewhere.'*

The velocity changes in the model domain suggest a local origin, although we cannot completely rule out advection from elsewhere. However, when combined with the variations in sea ice production interpolated from satellite observations between 2009 and 2012 (new Figure 2 below (Tamara et al., 2016), local production would appear likely. We have substantially rewritten the text to clarify this.

[Figure]

A. Cumulative sea ice prduction for 2009 (m yr⁻¹)    B. Cumulative sea ice prduction for 2012 (m yr⁻¹)

3. '*The results section includes discussion-type statements, beyond the results, which should appear in the discussion section. This confuses readers to judge what the data and model really show respectively, and what are author's arguments based on the observation and modelling.*'

We have addressed this in our resubmission, disentangling our observations from the model projections. We hope our rewrite clarifies this.

4. *In Section 3.2 (P6L18-20), sea ice production is presented as "modeled" but it is basically taken from Tamura et al. (2016) (P4L23-24). So, cumulative sea ice production presented in Figure 2A and 2B are somewhat observations, not really modelling outputs. I see a similar problem at P8L5.*

We apologise for any confusion. This was not our intention but rereading the text we can see this was ambiguous. We have now rewritten the text to clarify this point. We hope this change is satisfactory.

Specific comments.

1. *P2L24: remove one ")" and double "the"*
   We have corrected this in our resubmission
2. *P4L12: Why is it necessary to show the Microcat CTD data? The main body of the manuscript does not use the Microcat data at all, and it appears only in Supplement Figure S1.*
   The CTD data is essential as the XCTD's are working at their lower range (guaranteed to -2C), therefore direct comparison with a 'cold water' calibrated CTD provides confidence for us, and the reader, that the data is 100% reliable. We have stated this in the main text in our resubmission.
3. *P4L23: model/ocean coupling?*
   We have corrected this in our resubmission

4. *P5L24: Which depth range do you expect to see HSSW layer? It is said that the observed depth range is too shallow to observe it. The observed fact is the absence of HSSW, and inferring HSSW at greater depths is author's speculation in this case. Please more clearly distinguish observed facts and interpretations/discussion.*

We apologise. There is clearly confusion here: we are referring only to our observations in the Mertz NE sector, where pre-2010 HSSW was generally recorded between 400 and 500m; only one of our observations exceeds this depth (~550m). Crucially, both of our XCTD casts suggest a return of HSSW below 380m, with similar profiles to pre-2008 casts. So whilst we cannot say decisively that production of HSSW has resumed, the evidence is consistent with this interpretation. We hope our rewritten manuscript is clear in regards to this important point. Critically, this point has no bearing on our discussion of the developing polynya in the Commonwealth Bay NW region.

5. *P6L10: add a reference to support a statement on the absence of HSSW prior to the B09B grounding.*
We have added the relevant Laccrra et al. (2014) and Rintoul (1998) references here.

6. *P6L18: as the reviewer #2 pointed out, Tamura et al. (2016) is a boundary condition for the modelling, not a modelling output.*
We appreciate this and acknowledge it throughout our resubmission.

7. *P6L24-P7L1: It is confusing. Fig. 2C/D show that shallow water is warmer and fresher (red solid curves) than deep water, which is said immediate above. So, these statements are not consistent.*
We have addressed this in our resubmission.

8. *P8L8-10: Revise "The effect this change of…"*
We are not quite sure what this issue is in regards to this statement in our original draft?

9. *Figure 1:*
*- The caption can be clearer, starting for example "XCTD data. (a) locations of ....". It's better to say clearly what Figure 1 is about at the beginning, rather than what the panel a is about.*
We have addressed this for Figure 1 and the other figures in our resubmission

*- Blue labels on black parts of satellite imagery are very hard to read (especially on hard copy). Change the colour, or use a lighter colour to outline these letters.*
We have corrected this in our resubmission.

*- For panels (B)-(G), legends are common so it is adequate to show it in a single panel. In the text, the authors emphasized that salinity/temperature can be changed seasonally so data can be compared when they are collected in the same season. So, please indicate the month of these data together with the year; i.e. "December 201x" or such.*
Due to the variable depths our casts achieved and the range in observational values, reporting these figures as a single panel would be extremely difficult to interpret, and would lose key details our discussion highlights. We have, however, updated the months in the panel legend as requested.

*- It's hard to compare panels (B)-(G), because ordinate/abscissa of individual panels are not scaled. Are there any particular reasons not to scale these panels? Indeed, we need only one left axis for two panels next to each other. So, make only one left axis (not two left axes) and put one right axis to show approximate depths so that more readers can easily compare the observation and model results.*
We have now scaled the depth axes and updated this in our resubmission.

*- Panels (B) and (E) do not show the post calving 2012 data (green curves), and post calving 2011 data (black curves).*
This reflects the fact that no data is available for these sectors in these years.

*- Show the seafloor in panels (B) to (E).*
The bathymetry of the sea floor in this region is highly complex and highly variable. It would not be possible to place the sea floor on the figure and present the data at a suitable scale to observe the changes apparent in our observations when compared to previous years.

*- Many CTD positions are shown but only for the 2013. Can you show 2013 XCTD positions only if the data are presented in panels B-G and add previous data points that are currently shown in Figure S2? In other words, please consider presenting information in Figure 1a and Figure S1 in a single panel more efficiently.*
We have attempted this in our resubmission, however adding the previous year CTD station makes the figure too complicated, therefore we have updated both Figures to complement each other. We trust this is satisfactory but would welcome any further suggested changes.

10. *Figure 2:*
    *- Depth should be all positive numbers.*
    We have updated these in our resubmitted manuscript.
    *- I see two curves for each legend in the each panel; e.g. there are two blue solid curves in panel C. Why?*
    It is normal to run more than one model run to demonstrate the robust nature of the outputs. We have clarified this in point in our resubmission.
    *- Please add a short sentence at the beginning of the caption to emphasize that you show model results in Figure 2.*
    We have done this in our resubmission.
    *- Show the seafloor in panels (B) and (E).*
    Following our earlier point from Figure 1, the bathymetry of the sea floor in this region is highly complex and highly variable. It would not be possible to place the sea floor on the figure and present the data at a suitable scale to observe the changes apparent in our observations when compared to previous years.
    *Supplement P2L13: "fluxes"*
    *Thank you. We have updated this in the resubmission.*
    *Supplement P2L15: Fig. S3 is cited before Figs. S1 and S2 are cited. Please make sections in the supplement that are associated with Figures S1 and S2 so that the points of these figures are clearer. Showing just figures in supplement does not really help.*
    *We have updated this in the resubmission.*

As you remark, Reviewer Two's comments are of a more general nature but we have attempted to address all of their concerns. Many of the points are addressed above but the following provide specific details to the issues raised.

1. *The direct ocean observation and surface boundary condition (sea ice production estimated from satellite) support the hypothesis of high coastal polynya activities (i.e., high sea ice production and remnant signal of dense water) on the lee side of B9B.*

Our resubmission highlights the Taumura et al., (2016) sea ice reconstruction specifically to build upon the reviewer's point.

> 2. *The numerical model forced by the observed sea ice production perfectly fails to reproduce ocean properties before and after the MGT calving (Fig. 2c). Although authors insisted that the model vertical constant profiles in winter support the findings in observation, I think that the explanation does not convince readers (at least me). In my reading, modeling result does not support active dense water formation in the lee side of B09B.*

Dense water (HSSW) is defined as being colder than -1.5C and sometimes saltier than 34.5 psu) (Cougnon et al., 2013). Both our observations and modelling studies achieve these characteristics. Our modelling also suggests that under the post calving configuration this water is being formed locally, as opposed to advecting in from the Mertz and Commonwealth Bay Polynyas as it did pre-calving (Lecarra et al., 2014). Additionally, the observations based on the Taumra et al (2016) paper supports this interpretation, and whilst we cannot completely rule out advection from elsewhere, local production of HSSW would appear likely from our results. We have substantially rewritten the text to clarify this point specifically to address Reviewer Two's point here.

> 3. *Discussing polynya activity with depth average ocean flow in summer does not make any sense (see also my previous comment [4]). Strong baroclinic structure is developed in summer (e.g., Lacarra JGR 2014) and thus ocean velocity would strongly depend on depth. Moreover, I could not see where modeled HSSW exists.*

As outlined, we have updated the key model outputs to reflect this, and in our resubmission only use the lowest five bottom layers. Critically, this does not change our interpretation.

We thank the reviewers for their constructive comments. As a result, we feel the manuscript has been considerably improved. We remain excited by our results which we consider are highly topical and timely, given the importance of AABW formation to the global thermohaline circulation system. Critically, our study demonstrates that cryospheric changes, such as the grounding of B09B, can exhibit a first order control on HSSW and AABW formation. This observation may perhaps explain in-part the long-term variability in AABW formation as highlighted by the marine geological reconstructions. Thus we believe our findings are important and timely and thus eminently suitable as a Brief Communication for the broad multidisciplinary readership of *The Cryosphere*. We do hope you agree!

Best regards,

Dr Chris Fogwill on behalf of the co-authors

[revised manuscript text omitted]

**Contains:**
**Section 1: Model description**
**Section 2: Selection of model climatology**
**Figure S1: Comparison between XCTD and microcat salinities**
**Figure S2: CTD and XCTD station locations used in inter-comparison**
**Figure S3: Monthly surface heat (a) and salt (b) fluxes averaged over the Mertz Glacier Polynya (MGP) from Tamura et al., (2016) data set, with winter time averages (May to September inclusive) shown with crosses.**

**Section 1. Model description**

The model used here is based on the Rutgers version of the Regional Ocean Modeling System (ROMS) (Shchepetkin and McWilliams, 2005) that includes ocean/ice shelf and frazil ice thermo-dynamics (Galton-Fenzi et al., 2012; Dinniman et al., 2003). The horizontal and vertical grid is the same than presented in Cougnon et al. (2013̶6). Without a dynamic sea ice model, the fine-scale polynya activity is resolved by forcing the surface of the model with monthly heat and salt from Tamura et al. (2016) data set that is based on sea ice concentration estimated with the Tamura et al. (2007) algorithm. This algorithm estimates thin ice thickness using Special Sensor Microwave Imager (SSM/I) observations and the European Centre for Medium-Range Weather Forecast Re-Analysis data (ERA-Interim). Water masses formed on the continental shelf in the model are controlled by the variability of the air/sea forcing as well as by the glacial melt water released from the local ice shelves. The model has been set up to compare the ocean and basal ice shelf melting changes post-calving compared with other years of similar heat and salt fluxes intensity within the MGP region. The year 2009 and 2012 are chosen for the pre- and the post-calving air/sea forcing simulations respectively, after analysing the monthly heat and salt fluxes averaged over the Mertz polynyna area for the period 1992 to 2013 (Figure S3). The year 2009 is representative to an average to strong sea ice production year in terms of heat and salt fluxes and 2012 was chosen in consideration of the fast ice and its representation of permanent features between 2010 and 2012 (A. Fraser personal communication). Fast ice is parameterised as in Cougnon et al. (2013) and Cougnon (2016), using an updated version of Fraser et al. (2012). Lateral boundary fields, including salinity, potential temperature and horizontal velocity, were relaxed to a climatology calculated from the monthly fields from the Estimating Circulation and Climate of the Ocean, Phase II synthesis (ECCO2) for the period 1992-2013 (Wunsch et al., 2009). It is important to note that salinity values used in the model are on the Practical Salinity Scale (PSS78) and are dimensionless. The total run time of the model simulation was 33 years for each simulation. This 33 year run includes a spinup phase of 30 years to reach equilibrium using a repeating loop of the climatology forcing. A climatology of the last 3 years of the run are used for the analyses.

**Section 2. Selection of 2009 climatology**

The choice of the year 2009 for the PRE simulation forcing was made after analysing the monthly heat and salt fluxes averaged over the Mertz Glacier Polynya (MGP) area for the period 1992 to 2013. The period from 2007 to 2009 was identified as a constant sustained period with a winter average (May to September inclusive) of about -164 W m$^{-2}$, while the average over the pre-calving period (1992-2009) is of -159±17 W m$^{-2}$ (See Figure S3). Similarly, the salt fluxes averaged for 2007-2009 is of about 0.82 kg m$^{-2}$, while the averaged for 1992 to 2009 is of 0.82±0.1 kg m$^{-2}$. 2007 to 2009 can therefore be considered as being a representative period for the pre-calving MGP region. As a result, 2009 (the year closest to the calving) was chosen as the focus of the pre-calving simulation in this study to explore the general ocean conditions related to a stable ice geometry pre- and post-calving. Furthermore, given that only a single-year forcing is available for the post-calving simulation, a comparable single-year climatology is preferable for the pre-calving simulation. In the post-calving scenario, 2012 was chosen in consideration of the fast ice and its representation of permanent features between 2010 and

2012 (A. Fraser personal communication). In summary, the results from these simulations are not restricted to the year chosen for the forcing, they can be compared with other years of similar salt and heat flux intensity between 1992-2009.

[Figure]

[Figure]

[Figure]

**Figure S1. Comparison between XCTD and microcat temperatures (ºC) salinities (‰psu)**

[Figure]

**Figure S2.:** CTD and XCTD station locations used in inter-comparison

[Figure]

**Figure S3. Monthly surface heat (a) and salt (b) fluxes averaged over the Mertz Glacier Polynya (MGP) from Tamura et al., (2016) data set, with winter time averages (May to September inclusive) shown with crosses (Cougnon 2016).**

[Figure]

**Figure S4.**

Time averaged horizontal velocity at the bottom layer of the model (m s-1) for the pre-calving (red) and post-calving (blue) simulations, for extended Commonwealth bay/Mertz Polynya area. Note vastly decreased post-calving westwards flow from Mertz Polynya. The grey contours outline the ice mask used in the model for both simulations and the black contour outlines the coastline. The bathymetry of the model (m) is shown at the background.

---

## Author Response (AR3)

**BRIEF COMMUNICATION: IMPACTS OF A DEVELOPING POLYNYA OFF COMMONWEALTH BAY, EAST ANTARCTICA, TRIGGERED BY GROUNDING OF ICEBERG B09B**

Christopher J. Fogwill[1,2*], Erik van Sebille[3], Eva A. Cougnon[4,5,6], Chris S.M. Turney[2], Steve R. Rintoul[4,5,6], Benjamin K. Galton-Fenzi[5,7], Graeme F. Clark[2], E.M. Marzinelli[2], Eleanor B. Rainsley[8], Lionel Carter[9].

[1] PANGEA Research Centre, School of Biological, Earth and Environmental Sciences, UNSW Sydney, Australia

[2] Climate Change Research Centre, School of Biological, Earth and Environmental Sciences, UNSW Sydney, Australia

[3] Grantham Institute & Department of Physics, Imperial College London, United Kingdom

[4] Institute for Marine and Antarctic Studies, University of Tasmania, Private Bag 129, Hobart, Tasmania 7001, Australia.

[5] Antarctic Climate & Ecosystems Cooperative Research Centre, University of Tasmania, Private Bag 80, Hobart, Tasmania 7001

[6] Commonwealth Scientific and Industrial Research Organisation, Ocean and Atmospheric Research, Hobart, Australia.

[7] Australian Antarctic Division, Kingston, Tasmania

[8] Wollongong Isotope Geochronology Laboratory, School of Earth and Environmental Sciences, University of Wollongong, Wollongong, Australia.

[9] Antarctic Research Centre, Victoria University of Wellington, New Zealand

*Correspondence to c.fogwill@unsw.edu.au

**Response to reviewers**

We thank the reviewer for their further review, and in light of their comments and suggestions have updated our manuscript as outlined in the following paragraphs.

The reviewer raises four key points:

*(1) P2L6-7 "Supported by satellite observation and modelling, ..."*

*The expression "modeling" in the sentence is inadequate, because your modeling activity did not correctly represent changes of water mass before and after the calving event. Please consider to rephrase the sentence in the abstract, describing which part of your modeling support your conclusion/hypothesis (i.e., a signal of the modeled deep convection in the western side*

*of the B9B in the post-calving configuration)*

Whilst we understand the reviewer's point here, we believe the use of the word modelling is appropriate in the abstract, as the modelling has demonstrated the mechanism behind the deep convection and developing polynya that has begun in the lee of B09B, helping us make sense of the limited available observations (Figure 1). Whilst the 'water mass properties'

modelled in the simulations may not be precisely the same as those from the available observations, the observations and model simulations from Commonwealth Bay agree within within ~0.05‰ at 350m water depth. Perhaps, more importantly, as outlined, we modelled two *steady-state* configurations for the region: pre-calving, and an *equilibrated post-calving* state. The observations, conversely, are of an *unequilibrated* system in a state of change, and are not therefore directly comparable to the simulated water-mass properties. This said the model does allow us to study the broad scale circulation and HSSW

production well, both under pre and post calving configurations, thus it helps us understand the changes, making the use of the word 'modelling' crucial in the abstract. This can be clearly seen in Figure 3, and in the figure below (which we have added to the SI as Figure S5), which shows the bottom salinity within the Commonwealth Bay area averaged for winter (June, July, August), for A. pre- and B. post-calving simulations.

[Figure]

*(2) P9 L14-16 "... that our model simulation do not...for two stable ice geometries"*

*Authors explained a representation of modeled water properties in the pre-calving configuration in P8L18-P9L4. However,*

*the modeled salinity in the "NW" region was much higher than the observed one. In the 3rd review, I understood that the*

*representation in the pre-calving configuration was wrong/poor, and thus the model failed to represent the change of water*

*properties before and after the event. Therefore, the excuse in P9L14-16 (using only the two geometries) would not be a*

*reason for the failed simulation of the water mass change. I suggest that authors add several sentences to clearly describe*

*(2-1) What the model failed to reproduce water mass, i.e., water mass representation before the MGT calving event, and*

*(2-2) What you succeed in your model to support your conclusion/hypothesis, i.e., a signal deep convection in the lee side of*

*B9B.*

We have reiterated the limitations of the model simulations in our updated manuscript, but do not agree that the simulations either pre- or post-calving have 'failed' or are 'wrong', as outlined previously and in the manuscript the model captures the broad changes in circulation well under both configurations, and whilst it does not precisely reproduce the available limited observations, the modelling serves the purpose of demonstrating the changes in circulation, advection and impacts on HSSW water well. Furthermore, as addressed in the revised manuscript, we model two *steady-state* configurations for the region:

pre-calving, and an *equilibrated post-calving* state. The observations, conversely, are of an *unequilibrated* system in a state of change, and are not therefore directly comparable to the simulated water-mass properties. We hope that the additional caveats and supplementary figure demonstrate this.

*(3) Related to the comment (2-2), I suggest adding a map of bottom salinity or convection depth in winter in the post-calving configuration to show local dense water formation near the B9B. The figure would be helpful for reader to understand where the new dense shelf water is formed.*

This is a good suggestion, and we have added the figure to the SI as Figure S5.

*Minor comment:*

*(4) P8L25-P9L1 "Modelled water column ..."*
*This is a counter-intuitive expression.*

We have changed this sentence to: "Water column stratification in the model is stronger in winter when there is sea-ice production."

Again, we wish to thank the reviewers and the editor for their detailed comments, which have substantially strengthened this Brief Communication in the *Cryosphere*.

Dr Chris Fogwill on behalf of the co-authors.

[revised manuscript text omitted]

Figure 3. Results of high-resolution model simulations. Upper panels:  Simulated bottom current velocity (m/s) from ROMS, averaged on the 5 lowest layers of the model for both pre-calving (red vectors) and post-calving (cyan vectors) geometries near Commonwealth Bay for summer (A. November - December) and winter (B. August – September). The location of the Commonwealth Bay NW area is shown by a dashed white box and B09B is outlined in light grey. Lower panels: modelled salinity (C) and potential temperature (D) from independent simulations with ROMS (n=2), for the Commonwealth Bay NW for 'stable' pre (red) and post calving (blue) geometries, averaged for December (solid lines) and July (dashed lines).